# Basal-Type Breast Cancer Stem Cells Over-Express Chromosomal Passenger Complex Proteins

**DOI:** 10.3390/cells9030709

**Published:** 2020-03-13

**Authors:** Angela Schwarz-Cruz y Celis, Gisela Ceballos-Cancino, Karla Vazquez-Santillan, Magali Espinosa, Cecilia Zampedri, Ivan Bahena, Victor Ruiz, Vilma Maldonado, Jorge Melendez-Zajgla

**Affiliations:** 1Subdirection of Basic Research, Instituto Nacional de Medicina Genomica, Periférico Sur 4809, Col. Arenal Tepepan, Tlalpan, Mexico City 14610, Mexico; angelaschwarz.c@gmail.com (A.S.-C.y.C.); gceballos@inmegen.gob.mx (G.C.-C.); kivazquez@inmegen.gob.mx (K.V.-S.); mespinosa@inmegen.gob.mx (M.E.); mceciliazampedri@gmail.com (C.Z.); ivan.bahena@gmail.com (I.B.); vmaldonado@inmegen.gob.mx (V.M.); 2Present address Health Sciences departament, Universidad Autónoma Metropolitana, San Rafael Atlixco No. 186, Col. Vicentina, Iztapalapa 09340, Mexico; 3Instituto Nacional de Enfermedades Respiratorias. Calzada de Tlalpan 4502, Col. Sección XVI, Tlalpan, Mexico City 14080, Mexico; vicoruz@yahoo.com.mx

**Keywords:** breast cancer, networks, stem cells

## Abstract

(1) Aim: In the present paper we analyzed the transcriptome of CSCs (Cancer Stem Cells), in order to find defining molecular processes of breast cancer. (2) Methods: We performed RNA-Seq from CSCs isolated from the basal cell line MDA-MB-468. Enriched processes and networks were studied using the IPA (Ingenuity Pathway Analysis) tool. Validation was performed with qRT-PCR and the analysis of relevant genes was evaluated by overexpression, flow cytometry and in vivo zebrafish studies. Finally, the clinical relevance of these results was assessed using reported cohorts. (3) Results: We found that CSCs presented marked differences from the non-CSCs, including enrichment in transduction cascades related to stemness, cellular growth, proliferation and apoptosis. Interestingly, CSCs overexpressed a module of co-regulated Chromosomal Passenger Proteins including BIRC5 (survivin), INCENP and AURKB. Overexpression of BIRC5 increased the number of CSCs, as assessed by in vitro and in vivo zebrafish xenotransplant analyses. Analysis of previously published cohorts showed that this co-regulated module was not only overexpressed in basal breast tumors but also associated with relapse-free and overall survival in these patients. (4) Conclusions: These results underline the importance of Cancer Stem Cells in breast cancer progression and point toward the possible use of chromosomal passenger proteins as prognostic factors.

## 1. Introduction

Breast cancer is the most commonly diagnosed cancer among women from both developing and developed countries and it remains one of the leading causes of death in women worldwide [1]. Among the described molecular subtypes, patients with basal breast tumors present the poorest prognosis, with shorter metastasis-free and overall survival [2]. These tumors have a gene expression pattern very similar to mammary stem cells [3], and reciprocally, normal mammary stem cells overexpress a set of genes common to basal breast tumors [4]. Over the past few years Breast Cancer Stem Cells (BCSCs) have been proposed to be one of the cell populations responsible for this type of cancer’s heterogeneity, metastasis and tumor relapse. BCSCs have been isolated from tumor samples and breast cancer cell lines using a combination of surface markers such as CD44, CD24, EpCAM, CD133 and ALDH, among others [5]. Even with a large number of published studies that analyze the phenotypic properties of BCSCs, genome-wide gene expression studies are needed to gain new insights for a better understanding of the complexity of breast cancer biology. In the present report, we analyzed the transcriptome of BCSCs derived from the basal breast cancer cell line MDA-MB-468 in which we found enrichment in gene networks associated with several signaling cascades related to stemness. Among the regulated networks we found a novel module consisting of genes codifying for chromosomal passenger proteins, including BIRC5, INCENP and AURKB, which proved to be associated with relapse-free and overall survival in triple-negative breast tumors.

## 2. Materials and Methods

### 2.1. Cell Line

MDA-MB-468 basal breast cancer cells were purchased from the American Type Culture Collection (ATCC, MD). These cells were grown in Dulbecco’s Modified Minimal Medium (DMEM) supplemented with 5% Fetal Bovine Serum in a 37 °C and 5% CO_2_-humidified atmosphere.

### 2.2. Cancer Stem Cells Isolation

Cells were separated using a FACSAria BD cell sorter with the following cell surface markers: EpCam, CD44 and CD24, which have been reported previously [5]. The antibodies where bought from Miltenyi Biotec (Bergisch Gladbach, Germany) and BD Biosciences (Carlsbad, CA, USA). Briefly, cells were detached with Accutase (Life Technologies, Carlsbad, CA, USA) and washed with PBS 1%. A total of 15 × 10^6^ cells were resuspended in a stained buffer (PBS with FBS 1%) and then were incubated with antibodies for 30 min at 4 °C. FITC-IgG1 and PE-IgG1 (Miltenyi Biotec; Bergisch Gladbach, Germany) were used as isotypes controls. CSCs (CD44+/CD24low) and non-stem cells (CD44+/CD24+) were sorted on a FACSAria (fluorescence-activated cell sorter, Becton Dickinson, Franklin Lakes, NJ, USA). For validation, cells were seeded in MammoCult (StemCell Technologies; Vancouver, Canada) for 24 h before analyses. RNA was isolated using Trizol (Life Technologies, Mexico City, Mexico) and RNA was reverse transcribed with a Maxima First strand cDNA synthesis kit (Thermo Fisher Scientific, Rockford, IL, USA). The expression level of stem cell markers (ALDH1A1, OCT4, ALDH1A3 and KLF4) and survivin were assayed by real-time PCR using the SYBR or TaqMan Master Kit (Applied Biosystems, Mexico City, Mexico) and PCRs were performed on the QuantStudio 7 (Applied Biosystems).

### 2.3. RNA Analyses

Gene expression was analyzed by qRT-PCR using the double Delta-Ct method to compare the levels of transcripts. The sequence of primers and TaqMan assays are depicted in Appendix A. For the transcriptome assays, two pooled (two biological replicates each to a total of four) breast CSCs and non-CSCs RNA samples isolated from the MDA-MB-468 cell line were subjected to RNA-Seq analysis. Approximately 30–40 × 10^6^ 75 base-pair reads per sample were obtained. After quality control and trimming, reads were aligned against the GRCh38 human reference, using Gencode’s version 84 GTF and the STAR aligner [6]. Normalization and transcript quantification was performed using RNA-Seq by Expectation Maximization (RSEM) [7] and differential expression calculated with EBSeq [8]. Transcripts with an FDR < 0.05 were considered Differentially Expressed (DE). Relative expression was depicted as absolute fold change. Raw data were deposited at the NCBI SRA archive (accession number PRJNA509033. The networks, functional and upstream analyses were generated through the use of QIAGEN’s Ingenuity Pathway Analysis (IPA, QIAGEN Redwood City) and Gene Set Enrichment Analysis (GSEA) [9]. Correlation expression analysis was performed using COXPRESdb v6.0 [10], which aggregates 73,083 human microarray samples (Platform A-AFFY-44) from different tissues and experimental settings. Probe sets used were: 202095_s_at for BIRC5, 200853_at for H2AFZ, 209464_at for AURKB and 219769_at for INCENP. R coefficient correlation was plotted for all platforms. Since the analysis covered several platforms, COXPRESdb used supportability as a measure of the significance level, using the COXSIM values, as described in [10]. For this, the maxCOXSIM value derived from the Mutual Rank Score was compared with the null distribution generated under the same number of genes in the list of every platform employed. Expression levels of the chromosomal passenger proteins across breast cancer samples were obtained from the Gene expression-based Outcome for Breast Cancer Online (GOBO) tool [11], which includes 1881 breast samples from 11 public datasets. Expression levels for the analyzed genes in breast cancer subtypes were obtained from GEO (NCBI) and TCGA databases as normalized Fragments Per Kilobase Of Exon Per Million Fragments Mapped (FKPM; GSE58135 [12]) or Transcripts Per Kilobase Million (TPM; TCGA, database accessed 05/2016, [13]). The relapse-free and overall survival for the module was calculated using kmplot [14]. For this, the normalized value of the specific Affymetrix probe was used, the percentile between the lower and upper quartiles was computed and the best performing threshold was used as the final cutoff, as described in [15]. qRT-PCR validation: Stem and non-stem cells were isolated as described and seeded in MammoCult media (StemCell Technologies, BC, Canada) for 24 h.

### 2.4. Xenotransplant Assays

Wild type AB Zebrafish (*Danio rerio*; donated from Dr. Hilda Lomely from IBT-UNAM) were bred and maintained in standard conditions. Two days’ post-fertilization (dpf) embryos were dechorionated and anesthetized with tricaine (MS-222; Sigma, Mexico City, Mexico). Cells were isolated using Fluorescence-Activated Cell Sorting (FACS) or used directly. No tumors were observed in fibroblast-only injected animals. Embryos were microinjected in the middle of the embryonic yolk sac region at a final concentration of 37 cells for CSCs assays. For survivin-overexpressing assays 75, 125, 250 and 500 cells were injected. For each group of cells, 30 embryos were used, which were then monitored for five Days Post Injection (DPI). Only informative (e.g., surviving) embryos were assessed. In vivo limiting dilution assays were analyzed using the Extreme Limiting Dilution Assays (ELDA) software [16], which employs a generalized linear model used to compare active cell frequencies (stem cells) in cell populations. The project was revised and approved by INMEGEN’s scientific and ethical committees (09/2015/I).

### 2.5. BIRC5 Overexpression and Inhibition

To overexpress BIRC5 in MDA-MB-468 cells, the ORF of BIRC5 was cloned into the pQCXIP vector (Clontech, CA, USA). The construction was transiently transfected with Lipofectamine 2000 (Invitrogen, Mexico City, Mexico) and RNA was extracted 48 h after transfection. As a control an empty pQCXIP vector was used. To inhibit BIRC5 expression, the U6 promotor and a downstream 20 bp BIRC5-specific sequence (GACGACCCCATAGAGGAACA) were subcloned into the pQCXIP vector (Clontech). The construction was stably transfected in MDA-MB-468 cells. BIRC5 were decreased to approximately 75% (25% reduction) in these cells.

## 3. Results

### 3.1. Isolation of Breast Cancer Stem Cells

CSCs and non-CSCs were isolated using FACS from the triple negative breast cancer (TNBC) cell line MDA-MB-468 (Figure 1A). We used a combination of CD44, CD24 and EpCam markers to isolate CSCs since previous works have shown that these markers identify a subpopulation with increased tumor-initiating capabilities in triple-negative breast cancer cell lines [5] (Figure 1A). To verify that these surface markers indeed identified CSCs, we performed xenotransplant analyses in zebrafish. As expected, we found that CD44^+^/CD24^−/low^/EpCam^+^ MDA-MB-468 cells had higher tumor-initiating ability in this model (Figure 1B).

### 3.2. Breast Cancer Stem Cells Transcriptome Analysis

We next performed RNA-Seq analysis of sorted cells. We found 297 DE genes with a False Discovery Rate (FDR) < 0.05 (Figure 1C and Appendix A). As expected, these Differentially Expressed (DE) genes presented an overrepresentation of mammary-stem and, to a lesser extent tissue-stem-associated gene (FDR q-value 0.26), as assessed by GSEA (Figure 1D). Among them we found commonly stem-associated genes such as SOX9, Oct1 and Myc. In addition, a group of cytokeratins including KRT5, KRT6 and KRT14 were among the top 20 DE genes in the stem cell population (Figure 1E and Figure 2D). These proteins are part of the breast cancer’s molecular intrinsic gene expression classification that defines triple-negative breast tumors and are associated with worse prognosis [19]. As expected, qRT-PCR analyses validated the stem-related genes found to be upregulated in the RNASeq analyses (Figure 1F).

We then explored the implications of the DE genes found by performing network and pathway analysis using the IPA suite. As shown in Figure 2A, several canonical signaling pathways were predicted to be modulated in CSCs, including cascades previously reported as important regulators of breast cancer stem cell phenotype, such as the Death Receptor (*p* value: 1.08^−2^) [20], p53 (*p* value: 1.46^−2^) [21] and NF-kappaB (*p* value: 1.24^−2^) [22] pathways, among others. This coincides with enrichment in genes associated with cellular functions indispensable for the stem cell phenotype such as cell death, growth and cellular development (Figure 2B, *p*-value ranges: 7.41^−3^–1.73E^−8^, 6.12^−3^–3.50^−6^ and 6.46^−3^–3.50^−6^, respectively). Additional support for the enrichment in stem cell genes was found by performing a network analysis of DE mRNAs isoforms (after excluding DE genes), which also showed activation of cascades associated with stem cells (Appendix A). We then explored the upstream regulator genes that could explain the expression changes and cellular functions in CSCs. Figure 2C shows that several upstream regulators were inhibited, including interferon-related pathways (IFNL1, IFNA2 and IFNG), Prolactin, PAF1, EIF2AK2 and, as expected, CD24. Due to the sheer number of upstream regulators found, we performed an additional comprehensive “master regulator” analysis, which employs hierarchical clustering to point potential regulators that could be explaining the upstream signaling in these cells. The top hits of this analysis were ERBB2, Prolactin, HGF, BECN1, FAS and EPHA2 (*p* value of overlap: 8.17^−17^, 1.43^−16^, 1.69^−15^, 2.16^−15^ and 5.62^−14^, respectively). All these cascades have been described as important regulators of the stem cell phenotype in normal or malignant epithelial cells [23,24,25].

### 3.3. Genes from a Chromosomal Passenger Proteins Module (CPPM) are Overexpressed in CSCs

A more detailed analysis showed that one of the top general networks included a strongly induced expression module comprised by several basal cytokeratins (Figure 2D). As mentioned, these proteins are part of the breast cancer’s molecular intrinsic gene expression classification that defines triple-negative breast tumors [19], so we focused in it. Interestingly, in the same network we found deregulation of a Chromosomal Passenger Proteins Module (CPPM), which included BIRC5 (Survivin), INCENP, AURKB and H2AFZ. We validated the overexpression of BIRC5 and the stem cell marker ALDH1A3 by quantitative qRT-PCR (Figure 1F). BIRC5, INCENP and AURKB are three of the four proteins that constitute the chromosomal passenger complex (CPC), which is involved in maintaining a proper chromosome segregation and cytokinesis during mitosis (Figure 2D) [26]. To further support the existence of a CPPM co-regulated module in CSCs, we used a combined analysis of human gene expression datasets [10] and found that these proteins were positively co-expressed (BIRC5 Mutual Rank of co-expression of 14.4, 17 and 51.3 for AURKB, INCENP and H2AFZ, respectively; Figure 3A). Next, to explore the role of this module in the stem cell phenotype we overexpressed BIRC5 in MDA-MB-468 cells. Figure 3B shows that cells that overexpressed BIRC5 had a tendency to present higher expression levels of the stem markers ALDH1A1, NANOG and OCT4, although not statistically significant. As expected by these results, cells expressing BIRC5 showed an increased stem cell number, as assessed by xenotransplant dilution assays (Figure 3C–E). As expected, depletion of BIRC5 in MDA-MB-468 cells by an shRNA induced mitotic aberrations (20.01 aberrant mitosis per 10^6^ cells).

### 3.4. Genes from a CPPM Have Prognostic Significance in Breast Cancer Patients

Next, we analyzed the expression levels of this module in a previously reported breast cancer cohort [11] that includes 1881 breast cancer patients. All four genes showed higher expression levels in basal breast patients than the rest of the subtypes (Figure 4A) and, as expected, in ER negative tumors versus ER positive (Figure 4B). Concordantly, a higher level of the module’s composite expression was found in basal breast cancer patients (Figure 4C). It is important to note that basal breast tumors present stem expression signatures [3] and reciprocally, normal mammary stem cells overexpress a set of genes common to basal breast tumors [4]. Finally, we analyzed the prognostic significance of this gene expression module in a published set of 249 triple-negative breast cancer patients [14]. Figure 4D shows that basal breast tumor patients with overexpression of the CPPM had a worse prognosis (as assessed by relapse-free survival), than those without it (logrank *p* = 0.0052, hazard ratio = 1.45 (range 1.12–1.89)). This was not limited to basal breast tumor patients since overexpression of this module was associated with lower relapse-free or overall survival independent of the molecular subtype. (Figure 4E, logrank *p* = 1 × 10^−16^, hazard ratio = 1.94 (range 1.68–2.25), Figure 4F, logrank *p* = 1.4 × 10^−17^, hazard ratio = 1.89 (range 1.48–2.4) and Appendix A). As expected, the hazard ratios were higher and the *p* value lower in the analyses using the composite module than in the separated genes (Appendix A). These results support the importance of the CPPM found in CSCs on breast cancer progression.

## 4. Discussion

In the present report, we sought to analyze the transcriptome of CSCs derived from basal breast cancer cells. Uncovering CSC expression pattern will aid to the comprehension of their cell of origin as well as improve the breast cancer diagnosis and treatment. As expected, we found enrichment in genes associated not only with somatic tissue stem cells, but specifically to mammary stem cells. In addition, we also found cellular processes commonly associated to stemness such as cell death and survival, proliferation and development. These findings support the similarities between breast normal and tumoral stem cells. We were also able to find a number of putative upstream regulators that need to be further explored. In particular, we found a group of regulators that are part of the shared TGFβ and IL1 signaling, including IL1RN, MAP3K7, TAB1 and TGFβ1. This network plays a very important role in the responses elicited by environmental stress and, more important, it is able to induce a stem phenotype in breast cancer cells [20]. We also found a predicted decrease of several upstream regulators belonging to interferon signaling, including IFNL1, IFNA2, IFNG and Interferon alpha. It is interesting to note that IFN is able to inhibit breast cancer growth in in vitro and in vivo models [28,29], pointing toward the notion that this cytokine could be responsible for stem cells growth regulation. Indeed, we found that among the deregulated networks, cell growth was one of the most enriched GO functions. In our validation experiments, we isolated the stem population from MDA468 cells and seeded them in MammoCult ^TM^ media. MammoCult ^TM^ has been widely used in different types of cultures by many authors such as Choi HS et al. [30,31], Charpentier MS et al. [32] and Wolf J et al. [33], to evaluate and maintain breast cancer stem cells features while avoiding proliferation since it does not contain serum. Under these conditions, cells are not proliferating, so we cannot exclude that the enrichment in proliferation genes could be due to the cell culture conditions. Additional experiments using lineage tracing or in vivo visualization will be needed to address this result.

In addition to the signaling modules, we found the upregulation of a gene module consisting in BIRC5, AURKB, INCENP and H2AFZ. Three of these genes are part of the CPC, which is a key regulator and coordinator of several mitotic processes, such as activation of the spindle assembly checkpoint, correction of chromosome–microtubule attachment mistakes and regulation of the contractile structures that drives cytokinesis [26]. Chromosomal passenger proteins are the main molecular sensors of the spindle checkpoint [34] and thus, provide a key function to maintain ploidy in rapidly dividing cells. Since aneuploidy and the stemness phenotype are two of the main drivers of cancer progression [17], it has been proposed that these processes are tightly associated [18]. Indeed, modulating the expression of chromosomal passenger proteins, including BIRC5 and AURKB induces an increase in CSCs in various in vitro models [27,35]. In addition, it has been previously shown that BIRC5 is overexpressed in a subset of breast stem cancer cells [36,37]. These results support our findings.

Breast cancer is a heterogeneous disease in which molecular alterations, cellular composition, and clinical outcome provide a basis for diverse tumor classifications. In particular, Perou et al. identified four subtypes with clinical implications based on gene expression data: Luminal A (LumA), Luminal B (LumB), HER2-enriched and Basal-like [38]. In addition to classification, gene expression data has also been used as prognostic markers, as in the case of Oncotype DX® [39], a 21-gene signature that correlates with 10-year recurrence risk in ER-positive breast cancer. It is interesting to note that BIRC5 is one of the genes of the PAM50 signature (an intrinsic breast cancer subtype classifier) [40], and that high expression of this gene is one of the defining features of basal-like tumors, as defined by this approach. BIRC5 is also present in the Oncotype prognostic signature, which support its potential role as a potential prognostic marker. These findings are in concordance with the present results and help to provide a link between the expression of this gene and the main biological characteristic of basal breast cancer. Further analyses are required to establish if the CPPM could add additional value to the molecular classification of tumors, or, as we suggest, as a directed prognostic approach for basal-like breast tumors. Our finding that these proteins are co-regulated in CSCs further support this notion and points toward the presence of a still unknown common regulatory pathway that should be important in cancer progression, and an attractive therapeutic target. We also found that these proteins, alone or in combination, are potential prognostic factors, underlying their importance in cancer progression and providing a link between stemness and ploidy regulation, which needs to be explored.

Finally, it is important to note that in the present study, we only worked with a basal breast cancer cell line. Additional experiments using cell lines derived from all four breast cancer subtypes will be required in order to ascertain if our results are applicable to breast cancer of all molecular subtypes or exclusively to basal-like breast tumors.

## 5. Conclusions

In conclusion, we performed a whole-genome transcriptome analysis of breast cancer stem cells derived from a TNBC cell line, which showed an enrichment of transduction cascades related to stemness, proliferation and apoptosis. Interestingly, a CPPM was found to be overexpressed in cancer stem cells. This module was overexpressed in triple-negative breast cancer tumors and was associated with poor prognosis. These results underline the importance of BCSCs in breast cancer progression and point toward the possible use of chromosomal passenger proteins as prognostic factors.

## Figures and Tables

**Figure 1 cells-09-00709-f001:**
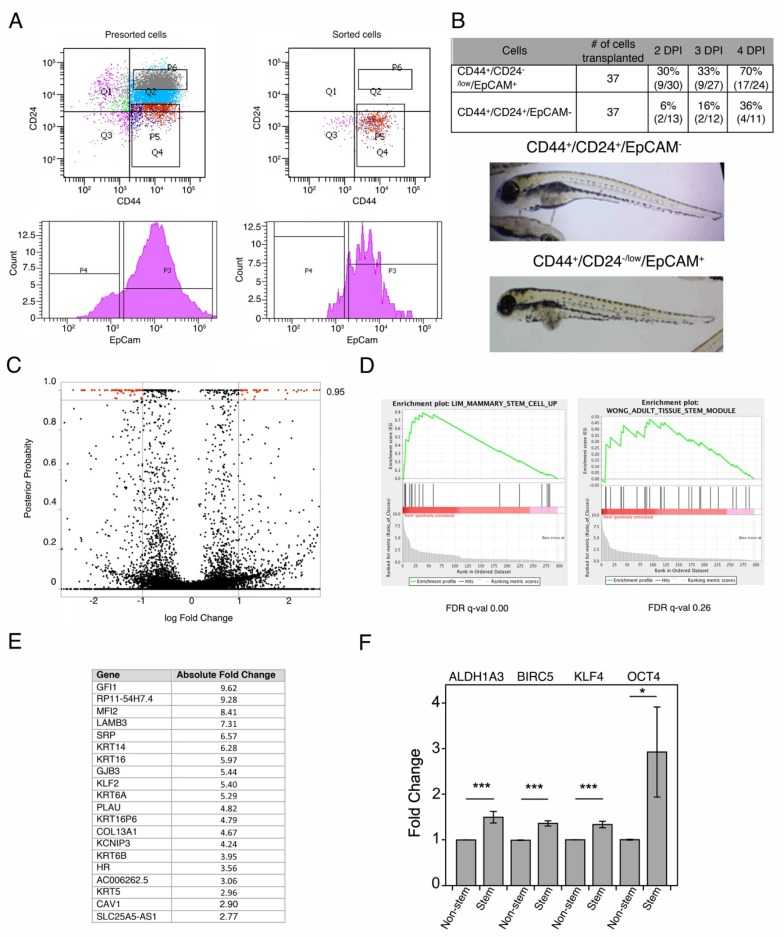
Isolation of Cancer Stem Cells. (**A**) Left panels: Flow cytometry analysis of MDA-MB-468 cells previous to sorting. The upper panel shows a dot plot with the fluorescence intensity of CD44 versus CD24 and the selected fields for sorting (CD44 high and CD24 low). The lower panel shows the third used marker (EpCam). Right panels: These plots display the purified population obtained after sorting. (**B**) Tumorigenicity assays using a zebrafish xenotransplant model. Upper panel: Tumor developed by MDA-MB-468 non-Cancer Stem Cells (CSCs; CD44^+^CD24^+^EpCAM^-^) and CSCs (CD44^+^CD24^−/low^EpCAM^+^) at 2, 3 and 4 DPI (Days Post Injection). For each experiment, 30 embryos were used and only informative (e.g., surviving) embryos were assessed. #: The number of cells transplanted to the embryos. Lower panel: Representative image taken at 2 DPI, showing abdominal tumors (arrows) in zebrafish embryos transplanted with MDA-MB-468 non-CSCs and MDA-MB-468 CSCs. (**C**) Volcano plot showing DE significant genes in red (posterior probability of being differentially expressed > 95%). The line shows the cutoff for significance. (**D**) GSEA analysis of the DE genes found. Left panel: Enrichment against a mammary stem cells signature (derived from [17]). Right panel: Enrichment against a tissue stem cell signature (derived from [18]). (**E**) Top 20 DE genes with a False Discovery Rate < 0.05. (**F**) Absolute fold change of representative genes in CD44^+^CD24^−/low^EpCAM^+^ (Stem) cells versus CD44^+^CD24^+^EpCAM^-^ (Non-stem) cells analyzed by qRT-PCR. Data derived from three independent FACS isolation experiments; * *p* < 0.05; *** *p* < 0.001.

**Figure 2 cells-09-00709-f002:**
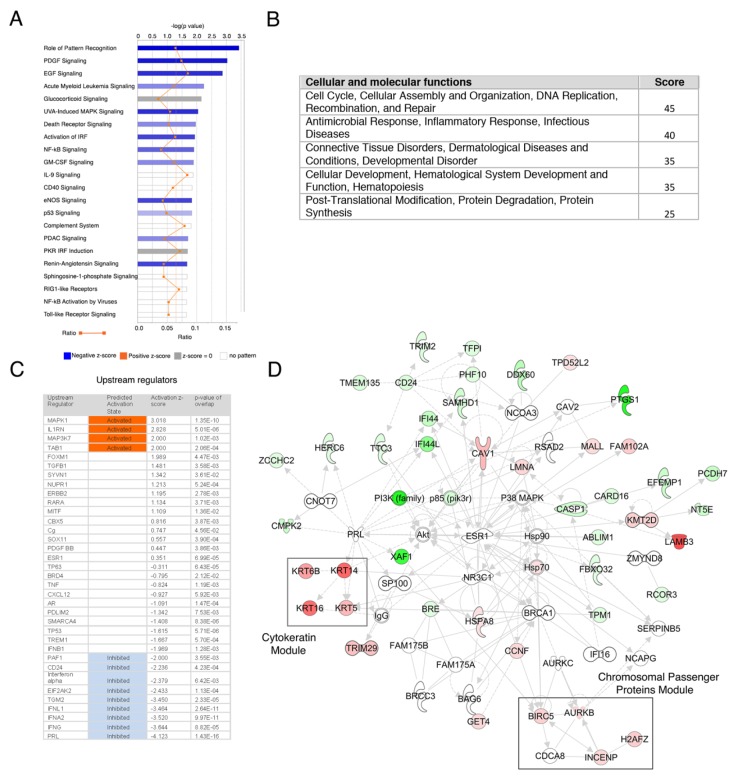
Network analysis of DE genes in CSCs from MDA-MB-468 cells. (**A**) Significant signaling cascades in the studied set. Bars length depicts the -log (*p* value) of each cascade, the color of the bar represents the z-score and the orange line points the ratio of genes in the pathway (e.g., the proportion of the studied DE genes present in each signaling cascade). (**B**) Enriched cellular and molecular functions table. The main functions and scores are shown. (**C**) Upstream regulators. The table shows the upstream regulator gene, the predicted activation (based on an absolute score > 2) and the *p* value of the gene overlap. (**D**) Main network in CSCs, showing two important modules in squares (Cytokeratin module and Chromosomal Passenger module). Red represents increased expression of the molecule in CSCs and green decreased expression.

**Figure 3 cells-09-00709-f003:**
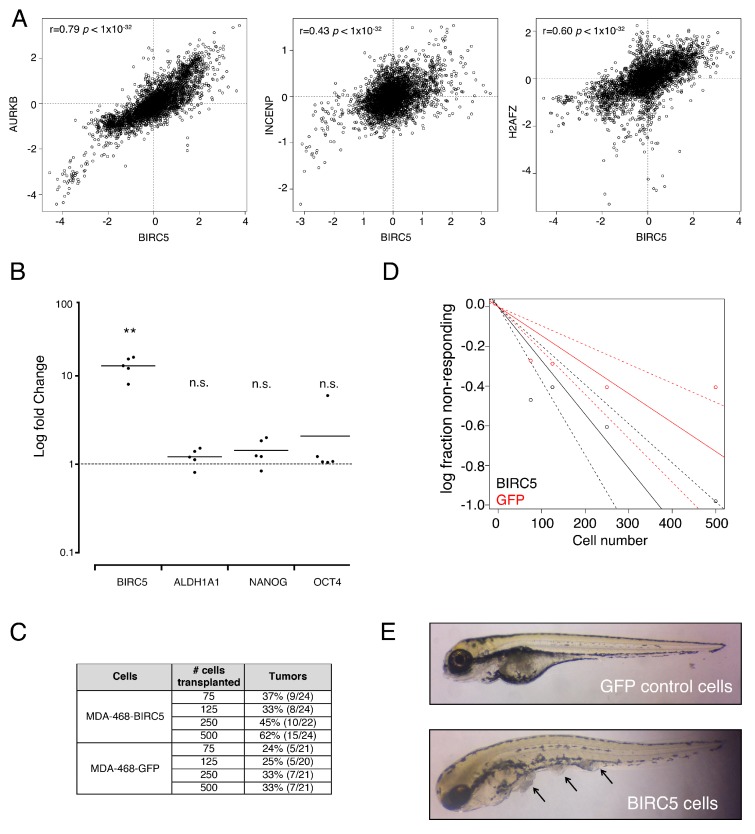
Participation of the Chromosomal Passenger Proteins Module (CPPM) in the stem cell phenotype of MDA-MB-468 cells. (**A**) Gene co-expression. Dot plots show the correlation between the expression of BIRC5 and AURKB (left panel), INCENP (middle panel) and H2AFZ (right panel). (**B**) MDA-MB-468 cells were transiently transfected with a vector containing a BIRC5 open reading frame and subjected to qRT-PCR for the genes shown. Log fold change refers to the logarithmic ratio between BIRC5-overexpressing cells versus empty vector-expressing cells [27], using the ∆∆Ct method, with SDHA as a housekeeping gene. Each dot represents an independent transfection. BIRC5 *p* = 0.0012, ALDH1A1 *p* = 0.1547, Nanog *p* = 0.1137 and Oct4 *p* = 0.3128 (** *p* < 0.01, n.s. = non-significant). (**C**) Xenotransplant dilution assays using a zebrafish model. Table showing tumor frequencies 4 DPI. # cells: number of cells injected. For each experiment, 30 embryos were used and only informative (e.g., surviving) embryos were assessed. Percentages and number of fishes with or without tumors are shown. MDA-468-BIRC5 cells overexpress BIRC5 and MDA-468-GFP overexpress a GFP transgene (control cells). (**D**) Limiting dilution analysis obtained with the Extreme Limiting Dilution Assays (ELDA) software. The plot shows the percentage of the embryos with abdominal tumors injected with MDA-MB-468 cells overexpressing BIRC5 or GFP-expressing (control) cells at 4 DPI. The difference between the analyzed groups was significant (***p* = 0.015; estimated stem cell frequency 1/369 vs. 1/685 in BIRC5- and GFP-expressing cells, respectively; n.s. non-significant). (**E**) Examples of tumors formed in these assays (arrows).

**Figure 4 cells-09-00709-f004:**
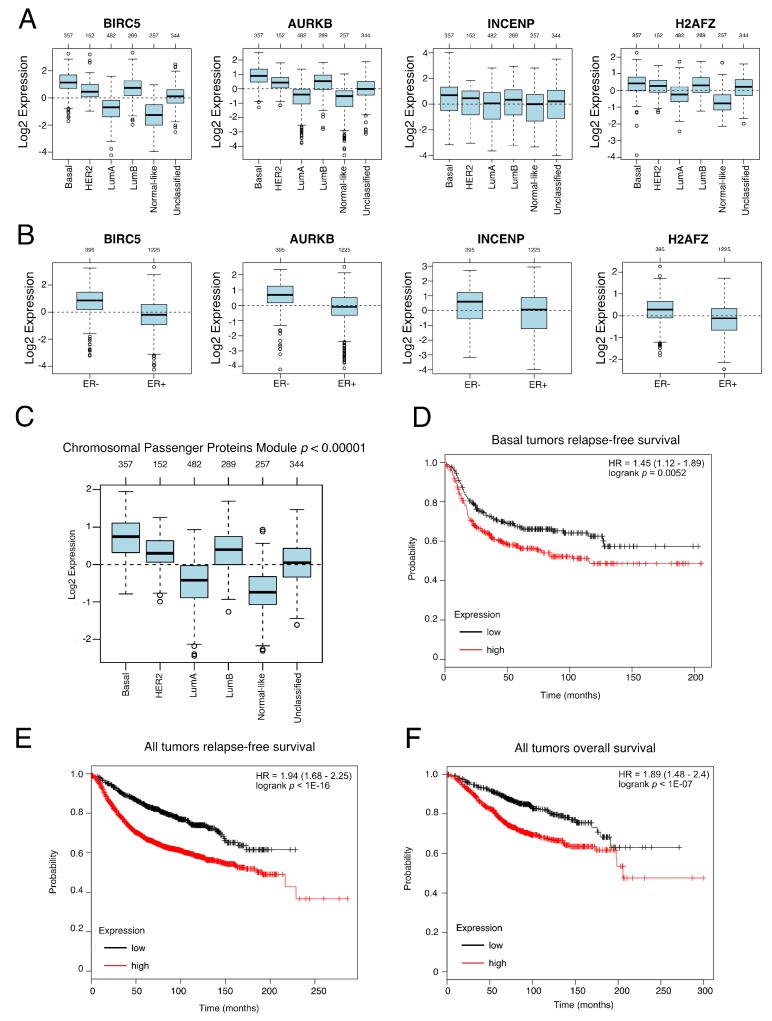
Clinical relevance of the CPPM in breast cancer. (**A**) Boxplot showing the normalized expression of BIRC5, AURKB, INCENP and H2AFZ in a group of 1881 breast cancer patients by breast cancer subtypes. The number of patients is shown above the plot. *p* < 0.00001 for all genes (ANOVA). (**B**) Boxplot showing the normalized expression of BIRC5, AURKB, INCENP and H2AFZ in a group of 1881 breast cancer patients by breast cancer ER status. The number of patients is shown above the plot. *p* < 0.00001 for all genes (ANOVA). (**C**) Boxplot showing normalized expression of the CPPM in a group of 1881 breast cancer patients. The number of patients is shown above the plot, *p* < 0.00001 (ANOVA). (**D**) Kaplan–Meier survival plot for relapse-free survival in patients with basal breast tumors derived from a published set of 249 triple-negative breast cancer patients [14], classified with the CPPM. High expression was defined as a mean expression of the module genes higher than 2093 logrank *p* = 0.0052, hazard ratio 1.45. (**E**) Kaplan–Meier survival plot for relapse-free survival in breast cancer patients derived from a published set of 249 triple-negative breast cancer patients [14], classified with the CPPM. High expression was defined as a mean expression of the module genes higher than 1223. logrank *p* = 1.10^−16^, hazard ratio 1.94. (**F**) Kaplan–Meier survival plot for overall survival in breast cancer patients derived from a published set of 249 triple-negative breast cancer patients [14], classified with the CPPM. High expression was defined as a mean expression of the module genes higher than 1454. logrank *p* = 1.10^−07^, hazard ratio 1.89.

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
