# Peer review of "Basal-Type Breast Cancer Stem Cells Over-Express Chromosomal Passenger Complex Proteins"

_cells, 2020, doi:10.3390/cells9030709_

Round 1

Reviewer 1 Report

Authors have sucessfully replied all my concerns, thus I strongly recommend to accept for publication the manuscript in its actual form.

Author Response

-Thank you for your comments. The article was thoroughly revised by a native English speaker.

Reviewer 2 Report

The work has been done with only cells derived from a single breast cancer cell line, MDA-MB-468. There are several types of breast cancer cell lines. Mainly ER+, EGFR+ and triple-negative tumors. To which one does MDA-MB-468 belong? This should be indicated in the text.

The data will be stronger if key results were validated with cells derived from another breast cancer cell line of the same type (basal).

The discussion should also indicate this restriction, and not generalized to all different types of breast cancer

Author Response

-Thank you for your comments. The article was thoroughly revised by a native English speaker.

The work has been done with only cells derived from a single breast cancer cell line, MDA-MB-468. There are several types of breast cancer cell lines. Mainly ER+, EGFR+ and triple-negative tumors. To which one does MDA-MB-468 belong? This should be indicated in the text.

The cell line is derived from a triple-negative breast cancer, and classified as a basal type (40. Neve RM, Chin K, Fridlyand J, et al. A collection of breast cancer cell lines for the study of functionally distinct cancer subtypes. Cancer Cell. 2006;10:515–527). This has been indicated in the text (lines 20, 49, 56, 268)

The data will be stronger if key results were validated with cells derived from another breast cancer cell line of the same type (basal).

We agree, but due to time restrictions, we are now unable to produce the required experiments. We discussed this limitation in the text.

The discussion should also indicate this restriction, and not generalized to all different types of breast cancer

We discussed the limitation in the text (line 322).

This manuscript is a resubmission of an earlier submission. The following is a list of the peer review reports and author responses from that submission.

Round 1

Reviewer 1 Report

MDA-MB-468 is triple-negative breast cancer cell line that also can expresses basal-like gene signature. MDA-MB-468 is model for studies on breast cancer biology and in vitro and in vivo metastasis assay. The authors showed that this cell line overexpresses the set of genes coding for Chromosomal Passenger 26 Proteins including BIRC5 (survivin), INCENP and AURKB.  Next, the explore the role of BIRC5 by expressing the gene in cancer stem cell-like cells selected by the markers EpCam, CD44 and CD24, and concluded could complete the stem cell phenotype observed in this cell lines.  Nonetheless, did not proved a correlation for the expression of BIRC5 and the stem markers ALDH1A1, NANOG and OCT4 in xenotransplants from the zebrafish assay. Overall the assays and methods used are well appropriated for this type of investigation. Discussion should be improved. Today, most studies have focused on the PAM-50 and 21-genes oncotype for clinical classification and selection of breast cancer patients for therapies.  We suggest the authors to revise and compare the set of genes identified their study and discuss on their importance and relevance on diagnostic and prognostic of subtypes of breast cancer as additional genes of PAM-50 or 21-gene signatures. Minor comment. Figure 4: Please identify the source of patients, specimens, DNA/RNA used in these assays.

Reviewer 2 Report

In this report, Schwarz and colleagues have studied the expression profile of some breast cancer stem cells. The study has been performed using only a single triple-negative breast cancer cell line (MDA-MB-468). The results was validated using xenotraplants in zebrafish and in cohort analysis of human breast cancer cases.

The work is well done, but there are some points that need to be considered and addressed.

The stem cell subpopulation was isolated based on “stem” markers by FARCS. However, it is important that despite its stem phenotype, these cells are growing. In tumors frequently, stem cells are not dividing and when they do it, it leads to the expansion or reappearance of the tumor.

The authors should include in some Figure a panel of proliferation markers of the cell line, in addition to the already present information on stem phenotype. It is likely that phospho-Rb is positive and cell cycle inhibitors are likely to be reduced or lost. This will be reflected in expression of genes such as BIRC5, INCENP and AURKB, all associated to mitotic progression.

The authors should determine what happens if these “stem” cells, isolated by FCAS, are placed in a culture media that permits survival, but not proliferation, such as serum deprivation or some other protocol. Stem cells may be dividing or not. The work is clearly performed on dividing cells with stem phenotype.

The discussion is very brief and mostly is a large summary of the results. The authors should modify it, and offer explanation about the meaning of their findings, and why they appear to be associated to mostly benign characteristics of tumors. BIRC5, INCENP and AURKB are proteins associated to proliferation and might reflect expansion of stem cells and not its malignant potential.

Reviewer 3 Report

Schwartz Cruz and coworkers present the manuscript entitled “Basal-type breast cancer stem cells over-express 2 chromosomal passenger complex proteins“. In general study was well conducted and provide novel data on stem cell phenotype on MDA-MB-468 cells. However, several concerns should be fully addressed before potential publication in Cells.

Materials and Methods

Please, better describe the methods and cohorts analyzed by Kaplan Meier. Data comes from public databases ?

Results

Figure 1. Breast Cancer Stem Cells Transcriptome analysis. The set of DE genes is interesting. However surprisingly no canonical genes associated to stem cells maintenance were identified. Instead of, a set of genes commons to any type of human cancer including Death Receptor, p53, and NF-kappaB, keratin, etc were found, as expected for breast cancer. Authors should better discuss these findings.

Figure 2. Breast Cancer Stem Cells Transcriptome analysis. QRT-PCR Validation of gene expression changes of a selected set of deregulated genes is missing and should be included in revised version.

Figure 3. Genes from a Chromosomal Passenger Proteins Module (CPPM) are overexpressed in CSCs. Although authors evaluated the Participation of the CPPM in the stem cell phenotype of MDA-MB-468 cells, no definitive data on the role of these proteins is showed. To define the role of deregulation of CCPM proteins in breast cancer stem cells, please evaluate the effects gene knockdown of a selected gene of the module, in cytokinesis or chromosome segregation of triple negative breast cancer cells.

Figure 4.  Clinical relevance of the CPPM in breast cancer. Details about the cohort data and clinical features showed in figure 4A-C and in 4D-E are missing and should be detailed here and in methods section. Figure 4E. Kaplan-Meier survival plot for 237 relapse-free survival in breast cancer patients classified with the CPPM. The analyzed was performed with the set of genes?, please show the survival analysis for each CPPM gene.

Discussion is poor and should be rewritten and enriched.